# Annealing Temperature Effects on Humidity Sensor Properties for Mg_0.5_W_0.5_Fe_2_O_4_ Spinel Ferrite

**DOI:** 10.3390/s22239182

**Published:** 2022-11-25

**Authors:** Iulian Petrila, Florin Tudorache

**Affiliations:** 1Faculty of Automatic Control and Computer Engineering, Gheorghe Asachi Technical University of Iasi, Str. Dimitrie Mangeron, No. 27, 700050 Iasi, Romania; 2Institute of Interdisciplinary Research, Department of Exact Science and Natural Sciences, Ramtech Center, Alexandru Ioan Cuza University of Iasi, Boulevard Carol I, No. 11, 700506 Iasi, Romania

**Keywords:** humidity sensors, annealing temperature, spinel structure, dielectric characterization, sensor material

## Abstract

The effects of annealing temperature on the structural, physical and humidity sensing properties of stoichiometric Mg_0.5_W_0.5_Fe_2_O_4_ spinel ferrite are investigated. In order to highlight the influence of sintering temperature on the structural, magnetic and electrical properties, ferrite samples were sintered for 2 h at 850 °C, 900 °C, 950 °C, 1000 °C and 1050 °C and the physical properties and humidity influence on magnesium-tungsten ferrite materials were analyzed. X-ray diffraction investigations confirmed the formation of magnesium-tungsten ferrite in the analyzed samples. SEM micrographs revealed the influence of annealing temperature on the microstructures of the samples and provided information related to their porosity and crystallite shape and size. This material, treated at different temperatures, is used as an active element in the construction of capacitive and resistive humidity sensors, whose characteristics were also investigated in order to determine the most suitable sintering temperature.

## 1. Introduction

A range of investigation approaches, e.g., theoretical, experimental and applicative, have been applied to ceramic materials due to their wide ranging applications [1,2,3,4,5]. Magnesium spinel ferrite is a remarkable type of ceramic material due to its properties, e.g., magnetization, coercivity, remanence, permeability, drug delivery and electrical conductivity [6,7,8]. One current goal of research groups is to find different ways to control and improve the dielectric parameters of this compound to make it much attractive from a non-magnetic perspective.

Magnesium ferrite with a spinel structure is one of the most important ceramic semiconductors. It has excellent electrical and magnetic properties and high chemical stability. Magnesium ferrite has a typical spinel structure in which the cation distribution in the crystal lattice site is sensitive to heat treatment due to the high diffusion of magnesium metal ions. Magnesium-based ferrite it is a very important ceramic material, with lot of practical applications, such as: magnetic recording media [9], transport properties [10], photocatalytic activity [11], humidity [12,13] and gas sensors [14,15,16], catalytic activity [17], optical, etc. [18]. This class of ferrite presents a particularly attractive set of properties: high Curie temperature, high electrical permittivity and resistivity and good chemical stability. Different addition to or substitution ratios with metal ions in the magnesium spinel ferrite stoichiometric composition can make essential contributions to the conduction mechanism, optical or magnetic properties of this semiconductor [19,20,21,22,23]. The main inconveniences for the synthesis of Mg-based ferrites are the high sintering temperature (over 850 °C) and longer time treatment, which result in significant power consumption and material loss [24]. One of the most widely encountered procedures to amend the properties of ferrites involves annealing at different temperatures and combining with another metal oxide, leading to different physical properties due to the heterogeneous interfaces that are created. By partially substituting magnesium metallic ions with tungsten metallic ions, a shorter sintering time and lower temperature can be used.

Ferrite materials can be used as active materials for humidity sensors because they offer good chemical and physical stability, notably due to their higher magnetic interactions compared to non-magnetic ceramic materials. Generally, ferrite compositions are investigated as materials for magnetic applications. Additionally, they have potential semiconductor applications in the fields of microelectronics and humidity or gas sensors. The conduction mechanism in ferrite is conditioned by factors such the composition, synthesis method, annealing conditions, porosity, milling time, defects and vacancies [25,26,27,28,29].

Because it is very easy to include tungsten metal ions in the structure of magnesium ferrite due to the multiple oxidation states of tungsten, the objective of this study is to provide information on the impact of annealing temperature on the structural, magnetic and electrical properties of magnesium-tungsten spinel ferrites. The goal of this research was to explore the possibility of using magnesium-tungsten spinel ferrite, annealed at various temperatures, as a sensitive material for humidity sensors. In the next paragraphs, we present the synthesis method, a structural analysis and the magnetic and dielectric properties, as well as discussing the behavior of magnesium-tungsten samples produced at different sintering temperatures, under humidity conditions.

## 2. Materials and Methods

The versatility of ferrite-based materials has made them applicable in various fields. Various substitutions or additions [30,31,32,33] of exotic metallic ions elements [34,35] have been made into the spinel ferrite structure in various synthesis methods such polymerized [36], hydrothermal [37], microwave combustion [38], molten salt method [39], sol-gel combustion [40] etc. Usually, spinel ferrites are produced using a standard ceramic technology method, which requires higher annealing temperatures and takes a long time for homogenization, yielding crystallites with small specific surface areas. The synthesis method plays a very significant role in determining the structural, magnetic and electrical properties of the resulting ferrite materials.

In order to obtain a material with good porosity and a high degree of homogeneity, as required for applications in the field of humidity sensors, we prepared a quantity of ferrite powder with the chemical formula Mg_0.5_W_0.5_Fe_2_O_4_ using sol-gel self-combustion technology [41]. The required reactants, having a high purity (99.9%) from Sigma-Aldrich, were weighed according to the following stoichiometric chemical formula: Mg(NO_3_)_2_·3 H_2_O, Fe(NO_3_)_3_·9 H_2_O and W_3_N_2_·3 H_2_O. Each of nitrates contains a mass concentration of 10% metal ions by volume of solution. We also used polyvinyl alcohol at a molar ratio of 1:1 with the total nitrate volume. The obtained homogeneous solution was neutralized to pH = 7 with ammonia at 10% concentration. In certain ratios and in a dry state, ammonia and polyvinyl alcohol can form pyrotechnic mixtures in which ammonia is the oxidant and polyvinyl alcohol is the combustion agent. As such, the mixture was dried, with care, under continuous observation at 100 °C for 60 min. The resulting powder of Mg_0.5_W_0.5_Fe_2_O_4_ ferrite was pre-calcined for 30 min at 500 °C. Using a Carver model 4350 hydraulic press with a force of 10 tones, the samples were uniaxially pressed into disks with a diameter of 6 mm and a thickness of about 1 mm and torus shapes with an internal diameter of 5 mm, an outer diameter of 13 mm and a thickness of about 2 mm.

To investigate the effects of annealing temperature on the structural, electrical and magnetic properties, the magnesium-tungsten ferrite samples were heat treated for 120 min using a heating rate of 3 °C/minute at the following temperatures: 850 °C, 900 °C, 950 °C, 1000 °C and 1050 °C.

Investigations of the structures of the samples were performed at room temperature using a Shimadzu LabX-6000 X-ray diffractometer and a TESCAN VEGA scanning electron microscope.

Electrical investigations, depending on the frequency and humidity, were carried out on both sides of the disk-shape samples, to which porous silver contact electrodes were applied. The electrical permittivity and resistivity, in a frequency range of 20 Hz–20 MHz, were also measured at room temperature (25 °C) using a Wayne Kerr 6400P impedance phase analyzer.

## 3. Results and Discussion

### 3.1. Structural Parameters

XRD diffraction patterns (Figure 1) for all ferrite series of samples were obtained at room temperature using a Shimadzu LabX XRD-6000 X-ray diffractometer with Cu-Kα radiation, λ = 1.5418 Å and a scanning speed of 2 degrees/minute.

Our analysis of the XRD spectra (see Figure 1) showed that 120 min treatment and an annealing temperature of 850 °C were enough to form a cubic spinel structure in the magnesium-tungsten ferrite. The peaks were in close agreement with the powder diffraction data from ICDD/JCPDS corresponding to cubic spinel ferrite structures (PDF 88-1943). Figure 1 shows that all samples mainly contained the Mg_0.5_W_0.5_Fe_2_O_4_ spinel phase, and the (311) peak displayed the strongest reflection. Increasing the annealing temperature diminished the quantity of Fe_2_O_3_. According to the literature on the substitution of Mg metallic ions with other metallic ions [42,43], it seems that tungsten metallic ions favor ferrite formation at an annealing temperature of below 1050 °C. The X-ray diffraction results of the magnesium-tungsten spinel ferrite, i.e., concerning reflections on lattice planes, are in accordance with other reported data on the preparation of magnesium ferrite [44,45,46,47].

The data shown in Table 1 allow us to conclude that in the case of magnesium-tungsten ferrite, increasing the annealing temperature also increases the bulk density, as well as the crystallite size and volume shrinkage. The bulk density results in the case of magnesium-tungsten spinel ferrite were similar with those reported in the literature [48,49,50].

SEM micrographs were obtained at room temperature by breaking the samples. It was found that the samples annealed at 850 °C and 900 °C for 120 min showed fine granulation, with crystallites of spherical shape and closed intra-granular porosity. These characteristics were confirmed by observed variations of relative permittivity and electrical resistivity based on relative humidity. As can be seen in Figure 2, the sample annealed at 950 °C showed clusters of micrometric crystallites with average size of 1.5 μm. These results were validated by the relative permittivity behavior of the sample, compared with that of the samples annealed at 850 °C and 1000 °C temperatures. The ferrite annealed at 1000 °C showed crystallites faceted with a higher average size, i.e., 2.7 μm, while the ferrite annealed at 1050 °C showed crystallites with an average size of 3.5 μm (see Figure 2).

We have shown that annealing at 1000 °C doubles the average size of the granules compared with annealing at 900 °C; therefore, the optimum temperature for the formation Mg_0.5_W_0.5_Fe_2_O_4_ ferrite, with crystallites with optimal density, is between 950 °C and 1100 °C. Similar results have been reported in the literature [51,52].

A comparative analysis of the SEM micrographs, shown in Figure 2, allows us to conclude that in the case of Mg_0.5_W_0.5_Fe_2_O_4_ ferrite, an increase in annealing temperature leads to modifications in the ferrite surface and the formation of a porous hollow structure. This morphology increases the specific surface area and improves to the sensing properties of the resulting ferrites.

### 3.2. Dielectric Investigations

Dielectric investigations of magnesium tungsten ferrite samples (disks) were performed at room temperature using a Wayne Kerr 6400P model impedance phase analyzer in a frequency range of 20 Hz–20 MHz. The relative permittivity and electrical resistivity were mainly dependent on the exchange mechanism between metallic ions located at tetrahedral or octahedral sites [53,54]. The frequency variation of permittivity and resistivity evidenced both the particularity of this exchange mechanism and the influence of crystallite distribution inside the network structure of the ferrite. Figure 3 summarizes the electrical characteristics of relative permittivity variation, determined in the absence of humidity with an alternating current regime in a frequency range 20 Hz–20 MHz, for magnesium-tungsten ferrite samples annealed at different temperatures. The electrical properties were strongly affected by the stoichiometry, impurities, porosity, annealing temperature and humidity. As can be seen in the graphical representation, the relative permittivity of the samples depended on the annealing temperature and crystallite size. Increasing the annealing temperature led to an increase in the relative permittivity of the samples. The sample annealed at 850 °C showed the minimum value of relative permittivity while the sample annealed at 1050 °C showed the maximum value. This was due to the fact that samples annealed at lower temperature have lower density, lower crystallite size etc. As can be seen in Figure 3, the relative permittivity of magnesium-tungsten ferrite is heavily dependent on frequency, showing a decrease of about two to three orders of magnitude for the frequency range 20 Hz–20 MHz, according with Maxwell-Wagner and Debye relaxation mechanism. Similar relative permittivity behavior has been reported in the literature [55,56,57,58].

The electrical resistivity of the analyzed samples is summarized in Figure 4. In this study, we found that electrical resistivity strongly depends on annealing temperature and frequency. In the case of the sample annealed at 850 °C, higher resistivity was observed compared with the sample annealed at 1050 °C. This phenomenon is due to the increase in the size of and the diminution of the space between the crystallites, which implies a significant contribution of the intrinsic conduction mechanism in ferrite materials.

Additionally, one observes that the electrical resistivity of samples is strongly frequency dependent. There was a noticeable decrease in electrical resistivity for all samples, i.e., by about two orders of magnitude, in the frequency range of 20 Hz–20 MHz. This is in line with results reported in the literature [59,60,61].

### 3.3. Magnetic Characterizations

Magnetic measurements were performed on the torus shaped samples at room temperature using an Wayne Kerr 6400P impedance phase analyzer at a frequency range 20 Hz–20 MHz. To evidence the impact of annealing temperature on magnesium-tungsten ferrite, we observed the evolution of two relevant magnetic parameters: relative permeability and Curie temperature. As can be seen in the graphical representation in Figure 5, the relative permeability depends on the annealing temperature. The minimum value of ferrite relative permeability was observed in the sample annealed at 850 °C, while the maximum value was found for the sample annealed at 1050 °C. From Figure 5, we can conclude that increasing the annealing temperature has the effect of increasing the relative permeability of the ferrite. Additionally, for the frequency range of 20–10^6^ Hz, we found that the relative permeability was constant for all samples. Similar results were reported in [62].

The Curie temperature (T_C_) of Mg_0.5_W_0.5_Fe_2_O_4_ ferrites was measured on the torus shaped samples, which were placed in a non-magnetic oven. As shown in Figure 6, the Curie temperature is slightly dependent on the annealing temperature.

We found that the sample annealed at 850 °C had the lowest value of Curie point, while the sample annealed at 1050 °C had the highest. The decrease in magnetic permeability was most dramatic in the sample annealed at 1050 °C. Based on our graphical representation, we can conclude the sample annealed at 1050 °C became paramagnetic at a higher temperature than the sample annealed at 850 °C. When the annealing temperature increased, we observed a slight shift of the Curie point toward the higher values.

### 3.4. Humidity Investigation

Humidity is a key factor influencing the behavior and electric characteristics of ferrites. The mechanism of humidity sensitivity for ferrites is based on the effect of water molecules, which are adsorbed on the material surface, changing the conductivity of the material. It is very significant to investigate how relative humidity influences the relative permittivity and electrical resistivity of ferrites. The influence of relative humidity on the Mg_0.5_W_0.5_Fe_2_O_4_ ferrite samples was measurement in a closed enclosure at a constant temperature of 25 °C using humidity values in the range 0–98% RH, as determined with an RLC meter at 1 kHz fixed frequency. The relative permittivity under the influence of humidity is summarized in Figure 7. As shown, the infiltration of water vapor into the ferrite has the effect of increasing the relative permittivity.

The resistive sensitivity of a humidity sensor based on Mg_0.5_W_0.5_Fe_2_O_4_ ferrite, depending on annealing temperature and humidity, is represented in Figure 8. The magnesium-tungsten spinel ferrite showed an electronic conduction mechanism due to humidity, which was reflected in the electrical resistivity (see Figure 8). In this study, we found that electrical resistivity was strongly dependent on the humidity level, with resistivity decreasing by approximately one order of magnitude when humidity increased from 0 to 98% RH.

The sample annealed at 1050 °C showed a greater decrease in electrical resistivity over the whole humidity range compared with the other samples. Still, we found, that decreasing the annealing temperature led to a decrease in electrical resistivity based on humidity. Although the variation and slope were small, samples annealed at lower temperatures were less sensitive to humidity. The sample annealed at 850 °C presented fine granulation with non-uniform closed pores, making its structure non-homogeneous (according to the SEM micrographs shown in Figure 2). The sample annealed at 1050 °C showed the highest sensitivity to humidity due to its open porosity and the size of the crystallites, which allow easy access of water vapor. The drop in electrical resistivity was most accentuated for the sample annealed at 1050 °C. Therefore, we can conclude that magnesium-tungsten ferrite annealed at high temperatures has a structure with open, well defined porosity distribution and good electrical resistivity response under the influence of humidity, which suggests the possibility of using it as a support material for humidity sensors.

The capacitive (Figure 9) and resistive type (Figure 10) sensitivity characteristics of magnesium-tungsten ferrite annealed at different temperatures were also investigated. As shown in Figure 9, for humidity levels of about 33%, the C/C_0_ coefficient indicated good sensitivity, while for humidity levels above 33%, a slight decrease in sensitivity was observed. However, in the case of capacitive sensors, the stability of the permittivity characteristics was not found entirely in the capacitive characteristics of the sensors, because the permittivity was measured on the massive material, while the capacitive sensors had other geometries and perturbative capacitive elements such as armatures, connectors etc. This is one of the reasons why capacitive characteristics are less well indicated than resistive ones for these types of sensors.

The C/C_0_ sensitivity coefficient was associated with annealing temperature, and by extension, to changes in the volume of open pores.

As shown in Figure 10, in the case of the ferrite sample treated at 1050 °C, the R/R_0_ coefficient showed the best sensitivity to humidity. This was attributed to the large number of open pores that were interconnected with each other, compared to those in the other samples. In the case of magnesium-tungsten ferrite, there was a clear correlation between annealing temperature and the degree of open porosity.

The investigated material showed good chemical and physical stability, with microstructures that are suitable for use in humidity sensors. The fact that it is a material with magnetic properties (ferrite) gives it additional advantages, both in terms of stability and application perspectives.

## 4. Conclusions

Mg_0.5_W_0.5_Fe_2_O_4_ ferrite was synthesized, as confirmed by XRD analysis, using sol-gel self-combustion technology. This approach is useful because it requires shorter times for annealing and lower production costs. An annealing temperature of at least 850 °C is required to achieve magnesium-tungsten ferrite spinel structures and crystallites with nanometric size.

The electrical and magnetic properties of Mg_0.5_W_0.5_Fe_2_O_4_ ferrite were found to be strongly dependent on the microstructure, porosity and annealing temperature. Therefore, for synthesis of ferrite, it is extremely important to carefully control the annealing temperature. The Curie temperature of the magnesium-tungsten ferrite slightly increased with increasing annealing temperature.

The variation of both relative permittivity and electrical resistivity were strongly depending on humidity. Annealing temperature plays an important role, having the effect of increasing the relative permittivity and decreasing the electrical resistivity. A significant number of open pores increased the electronic conduction in the presence of water vapor, indicating a high level of sensitivity to humidity.

The good humidity sensitivity observed with the Mg_0.5_W_0.5_Fe_2_O_4_ ferrite material annealed at higher temperatures indicates its suitability for use as a sensitive element for humidity sensors.

## Figures and Tables

**Figure 1 sensors-22-09182-f001:**
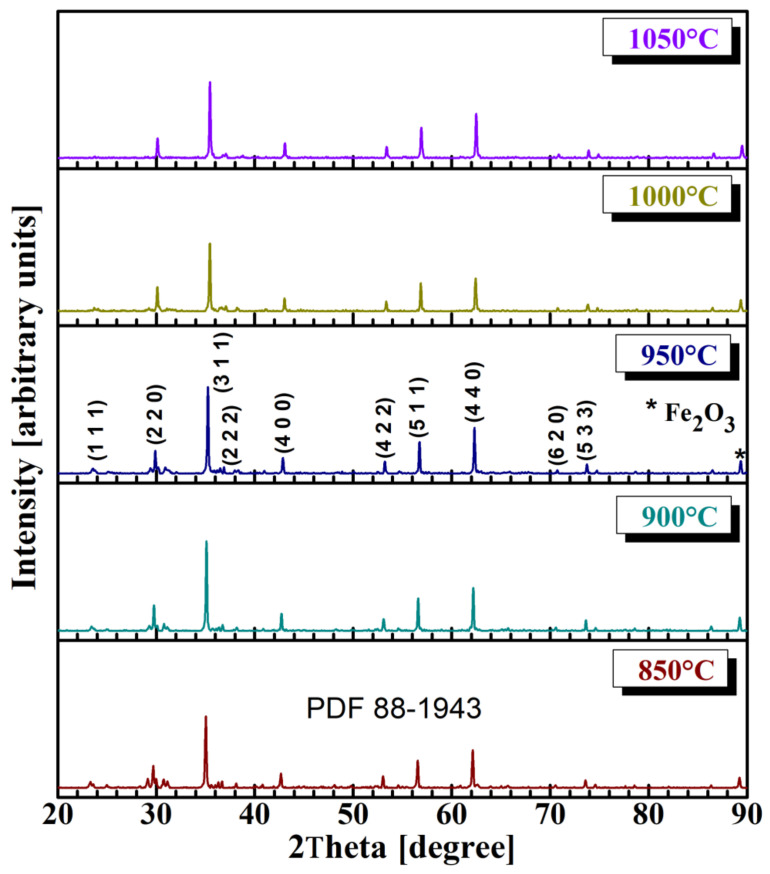
XRD characteristics of Mg_0.5_W_0.5_Fe_2_O_4_ powders annealed at various temperatures.

**Figure 2 sensors-22-09182-f002:**
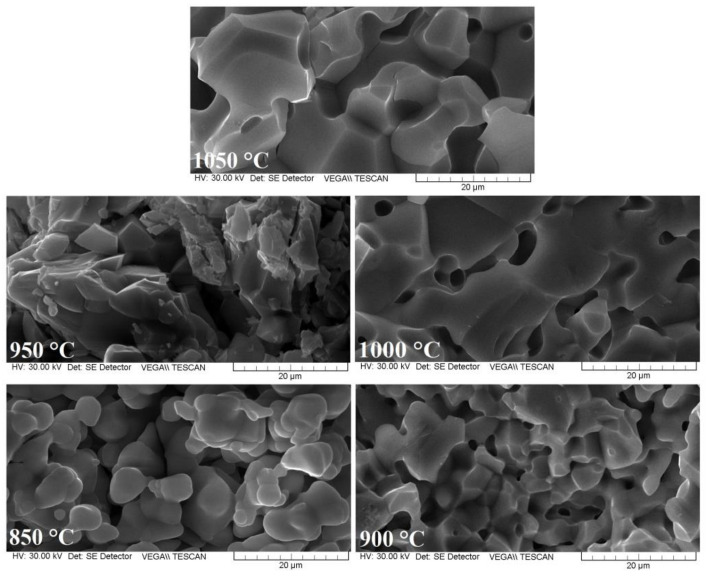
SEM micrographs of Mg_0.5_W_0.5_Fe_2_O_4_ annealed at various temperatures.

**Figure 3 sensors-22-09182-f003:**
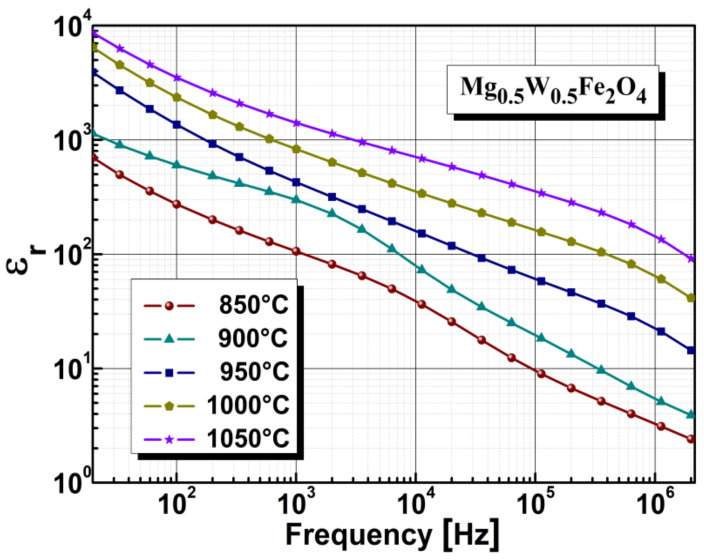
The effect of annealing temperature on the relative permittivity.

**Figure 4 sensors-22-09182-f004:**
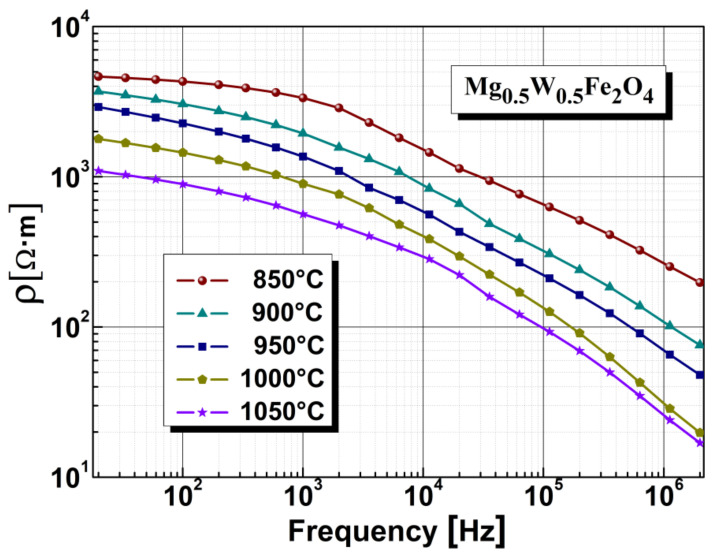
The influence of annealing temperature on electrical resistivity.

**Figure 5 sensors-22-09182-f005:**
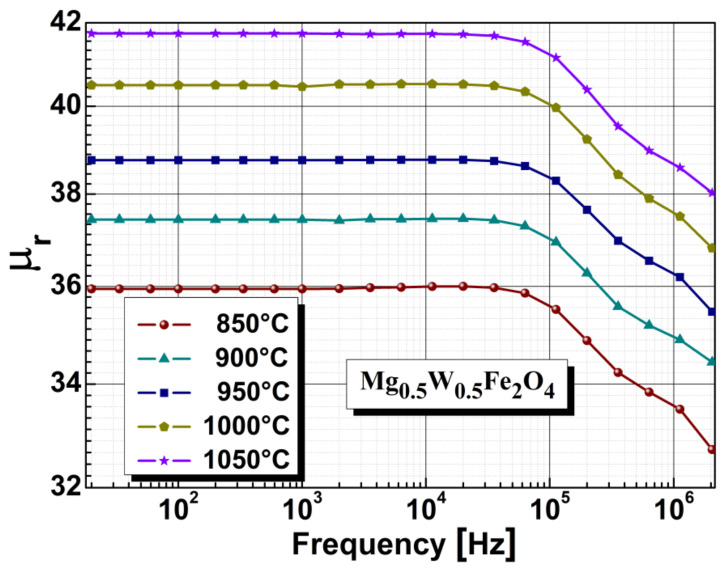
The influence of annealing temperature on the relative permeability characteristics of Mg_0.5_W_0.5_Fe_2_O_4_ ferrite.

**Figure 6 sensors-22-09182-f006:**
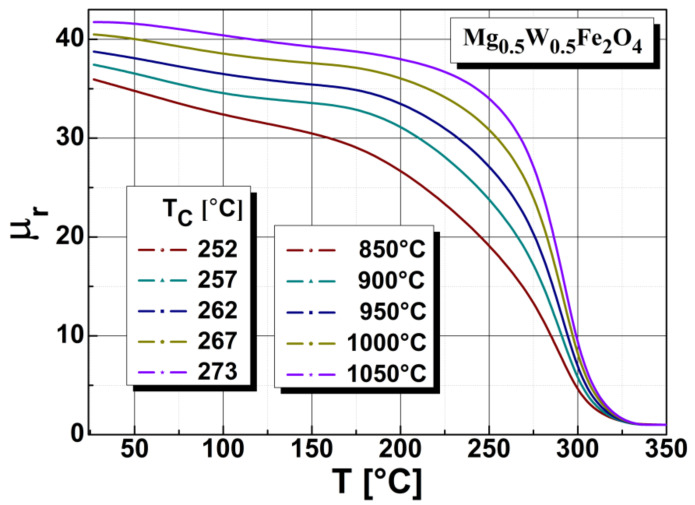
The influence of annealing temperature on the Curie temperature characteristics of Mg_0.5_W_0.5_Fe_2_O_4_ ferrite.

**Figure 7 sensors-22-09182-f007:**
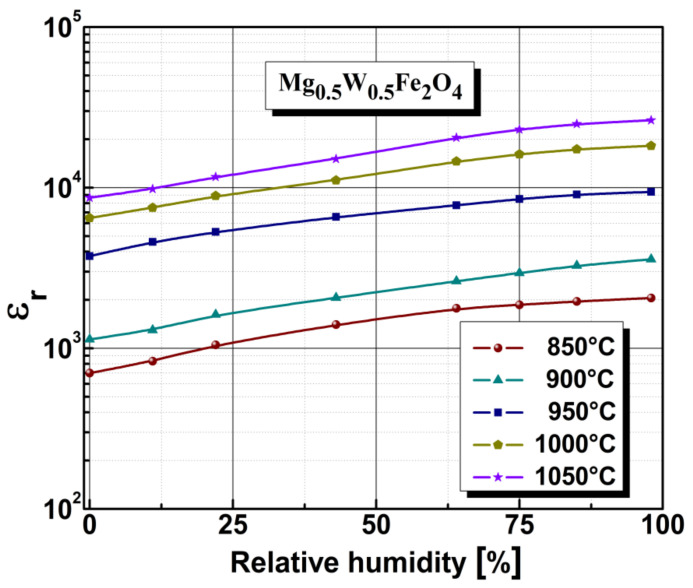
Relative permittivity under different humidity conditions for Mg_0.5_W_0.5_Fe_2_O_4_ ferrite.

**Figure 8 sensors-22-09182-f008:**
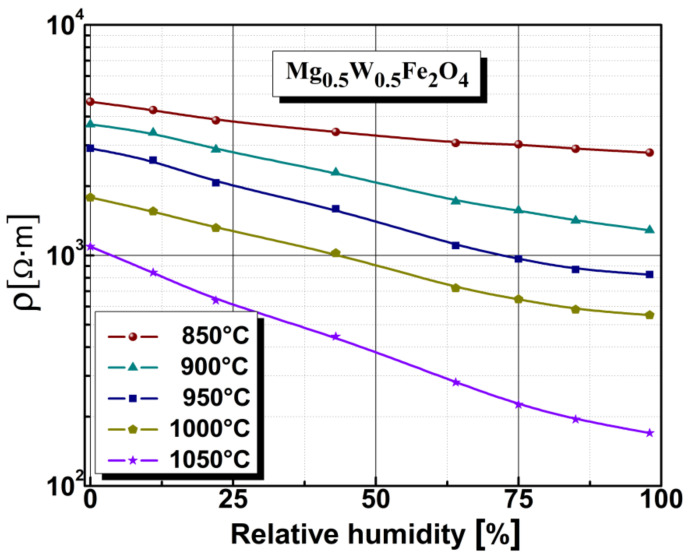
Electrical resistivity characteristics under humidity conditions for Mg_0.5_W_0.5_Fe_2_O_4_ ferrite.

**Figure 9 sensors-22-09182-f009:**
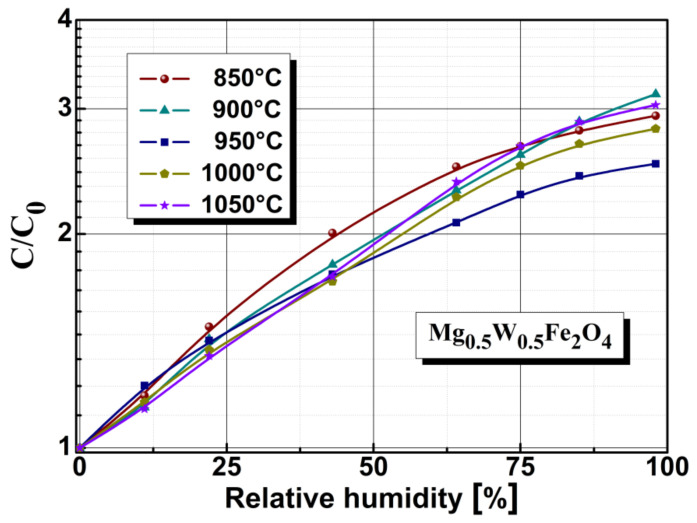
Electrical capacitance of sensors based on Mg_0.5_W_0.5_Fe_2_O_4_ ferrite material annealed at various temperatures.

**Figure 10 sensors-22-09182-f010:**
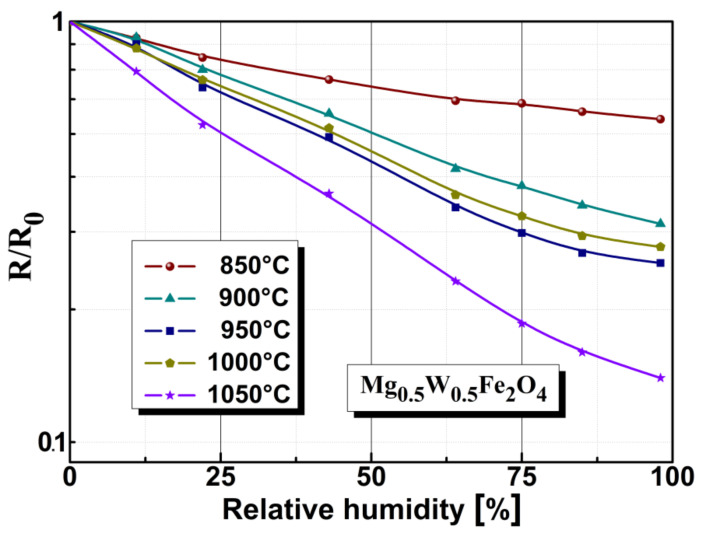
Electrical of sensors based on Mg_0.5_W_0.5_Fe_2_O_4_ ferrite material annealed at various temperatures.

**Table 1 sensors-22-09182-t001:** Structural characteristics of Mg_0.5_W_0.5_Fe_2_O_4_ ferrite samples.

T_annealing_ [°C]	Bulk Density [g/cm^3^]	Volume Shrinkage ΔV/V [%]	Crystallite Size D [μm]	Ferrite Purity [%]
**850**	4.82	45.8	0.6	94.5
**900**	4.92	47.3	0.9	94.8
**950**	5.17	48.8	1.5	95.3
**1000**	5.27	49.4	2.7	95.9
**1050**	5.36	50.1	3.5	96.1

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
