# Peer review of "Annealing Temperature Effects on Humidity Sensor Properties for Mg0.5W0.5Fe2O4 Spinel Ferrite"

_sensors, 2022, doi:10.3390/s22239182_

Round 1

Reviewer 1 Report

This manuscript studies the annealing temperature effects of Mg0.5W0.5Fe2O4 spinel ferrite on the electric and magnetic permeability, which shows potential applications on humidity sensors. I have the following comments:

(1)      The part that needs significant improvements is that: the contribution and improvements of this work is quite unclear compared to previous studies. On one hand, whether other elements have been tried in the formation of magnesium ferrite, how is the performance compared to this study. On the other hand, the humidly sensors based various principles have been widely utilized. This study has investigated the variation of permittivity and resistivity with the humidity, how is the result compared to the state of art?

(2)      This study has varied the annealing temperature from 850 degrees to 1050 degrees, the permittivity is the smallest at 850 degrees, while the resistivity and permeability increase with the annealing temperature. On one hand, further explanations and discussions are needed from the perspective of solid state physics and materials disciplines  to discuss the varying trend . On the other hand, other temperatures smaller than 850 degrees and larger than 1050 degrees could be optimal, which seems quite possible according to this study. Why the authors have not tried other annealing temperatures?

(3)      For Fig.9, the variations of capacitance with annealing temperature are not monotonous, which seems to contradict with the variation trend of permittivity. Please try to add some discussions.

(4)       Whether the variations of electric and magnetic permeability can be integrated when designing the humidity sensor?

Reviewer 2 Report

The authors have reported the annealing temperature effects on microstructure, relative permittivity, and resistivity, relative permeability of Mg0.5W0.5Fe2O4 ferrite, which are sensitive to the humidity and can be used as humidity sensor. However, there are some major problems in this manuscript. First, the manuscript is not organized well, especially the experimental and conclusion section. Second, only a few sentences can be viewed as one paragraph in text. Third, some discussions on mechanism are not clear, such as the influences of the annealing temperature on the relative permittivity, grain size, and so on. After carefully reading the manuscript, my suggestion is rejection. All the comments are listed as followed.

1.       In section of Materials and methods, the paragraph from line 65 to 74 can be deleted, which can transfer to the introduction. By the way, the text from line 93 to 104 can be combined into one paragraph. Similarly, in the section of results and discussions, “The XRD diffraction patterns (Figure 1) for all ferrite series of samples, were investigated at room temperature using X-ray diffraction through a Shimadzu LabX XRD-6000 diffractometer using Cu-Kα radiation with (λ = 1.5418 Å) and a scanning speed of 2 degrees/minute.”, “The dielectric investigations of magnesium tungsten ferrite samples was performed at room temperature, on the disks, using an Impedance Phase Analyzer Wayne Kerr 6400P model, for the frequency range 20 Hz ÷ 20 MHz”, all which can be added into the experimental part.

2.       Some mistakes, such as “20 Hz ÷ 20 MHz” and “0 ÷ 98 %RH”.

3.       In Figure 1, the PDF 88-1943 should be added into the figure for comparison

4.       In Table 1, the volume shrinkage can be derived from the density. Meanwhile, how to calculate the ferrite purity?

5.       According to SEM, it is difficult to observe the grain size. Larger-area SEM iamge should be used.

6.       In Figure 3, “The explanation consist in the fact that in case of sample annealed at 850 °C it has a higher number of closed pores compared to the sample annealed at 1050 °C.” This explanation is not accurate. There is one equation which can give the relationship between the porosity and the relative permittivity.

7.       In Figure 6, it is difficult to obtain the Curie temperature. Or how to define them?

8.       In references, some mistakes, such as “Sens. Actuators B Chem. 2012, 171– 172, 832– 837”.

Reviewer 3 Report

1. In table 1 crystalline size may be written in terms of nanomters (nm). This will help properties of materials at small level

2. In results section, properties of ferrite materials also found to depends in low temperature- may be added. For this following reference may be added 

Reference- R K Verma, R K Singh, A Narayan,  L Verma, A K Singh, A Kumar, JTAC, Low temperature temperature synthesis of hexagonl barium hexaferrite by annealing temperature at 450C followed by quenching, 2017, p.691-699.

Reviewer 4 Report

The overall of the work presented its interesting and excited regardless to some grammatical mistakes. Further, I have summarized my feedback on this impressive research paper are as following:

1.     There are some references I think are not highly related so, the author needs to justify or remove/replaced with other one more relevant. These references such as:

a)     Reference no “1”

b)     Reference no “3”

c)     Reference no “5”

d)     Reference no “7”

e)     Reference no “12”

f)      Reference no “27”

2.     The obtained results of this research work are explained, and it seems like a good result but somehow, I did not find any validation study as evident to convince a reviewer.

Round 2

Reviewer 1 Report

Thanks for  the revision. 

Reviewer 2 Report

It can be accepted in current form.